# Complications Arising from Dental Trauma Incurred from Falls Involving Geriatric Patients: A Case Report

**DOI:** 10.3390/diagnostics13193021

**Published:** 2023-09-22

**Authors:** Sebastiana Arroyo Bote, Catalina Bennasar Verges, David Ribas-Perez, Diego Rodriguez Menacho, Paloma Villalva Hernandez-Franch, Ignacio Barbero Navarro, Antonio Castaño Séiquer

**Affiliations:** 1University of the Balearic Islands, 07006 Palma de Mallorca, Spain; s.arroyo@eua.edu.es (S.A.B.);; 2Department of Stomatology, University of Seville, 41001 Seville, Spain; 3Andalusian Health Service, 36420 Andalucia, Spain

**Keywords:** crown fracture, dental trauma, falls in the elderly, soft tissues, clinical examination

## Abstract

Elderly patients have a high incidence of falls that result in facial injuries. Many cases present with dental crown fractures and soft-tissue lacerations. The initial diagnosis is especially important in this type of dental trauma, since an initial error can result in the failure to establish the correct treatment from the very beginning, worsening the prognosis, which, in the worst case, can compromise the teeth affected by the trauma. Case report: We present the case of a patient, a 79-year-old woman, who suffered an accidental fall. The patient was examined by a dentist in the emergency room. She was diagnosed with a simple crown fracture of the right upper central incisor and right upper lateral incisor without any pulpal involvement and with laceration of the lower lip. Months later, she had an episode of inflammation in the lower lip, and she developed edema and pain. An X-ray of the lower lip revealed a radiopaque mass compatible with a fragment of dental tissue due to the inclusion of remnants of the dental tissue resulting from the previous accident fall. Conclusions: The population of geriatric patients is steadily increasing in our society, and a higher frequency of falls has been observed in this age group, affecting the maxillofacial region, with consequent dental trauma on many occasions. Therefore, as professionals, we must pay special attention to the prevention and treatment of this problem, insisting on the importance of routine investigation in the emergency visit for dental trauma, which includes a meticulous examination of the soft tissues accompanied by a radiographic examination, when the inclusion of foreign bodies is suspected.

## 1. Introduction

The World Health Organization (WHO) considers that the incidence of traumatic dental injury (TDI) varying throughout the world according to region, the lack of standardization in the studies of incidence and prevalence, and the different cultures and habits inherent in any given population all make the collection of accurate data on TDIs very difficult. Nevertheless, Petti et al. [1] estimate that there are currently a million people in the world who have experienced a TDI, which highlights the importance of this health issue and shows that it is on a par with acute/chronic illnesses that affect the human race.

Such traumas are the result of accidents that can occur in any area of our lives: the home, the workplace, during sport, using transport, etc. In the event of any traumatic incident, it is vital that the injured party receive appropriate first aid. If correct measures are taken quickly enough, ideally within the first few minutes, the consequences of the injury and the period of recovery can be greatly reduced, thereby limiting the impact on the patient’s life.

Every episode of dental trauma has its own characteristics. A simple injury that may initially seem innocuous can present complications over time and therefore have a significant impact on the patient’s quality of life. Quality of life includes physical, psychological, and social well-being. For some years now, different indicators have been developed to measure the impact of certain oral conditions, such as TDI, that affect people’s quality of life. A lived experience such as an TDI can affect social and emotional well-being as well as oral functions. Certain common daily situations, such as smiling, laughing openly, showing teeth without hesitation, maintaining a non-irritable emotional state, or eating or enjoying food, can be altered as a result of an TDI, hence the importance of an early diagnosis in order to develop and implement an accurate and precise treatment plan from the outset [2,3].

The most common dental trauma in definitive dentition is a coronary fracture, especially in the upper incisors. Fractures involving hard tissue usually involve damage to areas of perioral soft tissue; 62.8% of patients requiring treatment in emergency centers have perioral soft tissue damage [4]. The groups who suffer the most frequent dental trauma are infants and adolescents, but, nowadays, the rise in life expectancy has resulted in increased activity by one part of the population with advanced age, and on the other hand, the instability and movement dependency in another part of the geriatrics patients make this group particularly vulnerable to injury to maxillofacial areas as a result of falling [5,6,7,8], although physical abuse and traffic accidents also contribute to facial injuries in this group [6,8,9,10]. According to studies conducted by Müller et al. [11], 42% of elderly patients seen in emergency units were injured as the result of a fall, with the larger proportion being women [7]. A third of people over the age of 65 fall every year, and half of them fall at least twice a year.

The world’s standard population is undergoing substantial changes as a result of declining mortality, resulting in a rise in the population’s life expectancy. The aging process in humans is physiological, and it is associated with a number of cellular and metabolic abnormalities. Such loss of physical functionality happens gradually, rendering the aged more prone to developing many illnesses and lowering their quality of life [12].

With regard to the physiological intrinsic features of the aging population’s body, such as anatomical abnormalities that are functional and impacted by disease and chronic drug use, they have been one of the pathologies disabling the trauma [13].

Residents in care homes fall three times more than people who live in their own homes. The risk factors are inadequate physical exercise, insufficient motor skills, high blood pressure, and diabetes. Even though the majority of falls do not result in serious injury, between 10% and 25% result in body trauma [14].

TDIs are present in 85% of accidents that involve the mouth region, which is 5% of injuries suffered at any age. We have found no studies relating to the incidence of dental damage in geriatric medicine, but the European multicenter study [15] on injuries to maxillofacial areas in patients aged 70 years and older highlights the significance of facial trauma suffered by a total of 1334 patients over a five-year period. The most frequent fracture seen was to the zygomatic arch (515), followed by a mandibular fracture (414). No dental fractures were reported in this study, but on many occasions, facial injuries have been accompanied by dental trauma. Recent studies [16] have shown the importance of preventative and adequate treatment in the elderly (who form part of this group) with special needs—either medical or psychological—and the need for detailed, instructive courses for professionals.

Patients who are older need the medical team providing them with care to be more prepared since they are more likely to experience serious problems, surgical morbidity, expensive medical care, and hospital stays [13,17].

In this type of patient and in the case of trauma involving the teeth, for a correct diagnosis, the International Association of Dental Traumatology (IADT) has developed the so-called Trauma Pathfinder, which establishes a simple path to achieve a correct diagnosis of the traumatized teeth [18]. This diagnostic path begins with the identification of the existence of dental luxations. Subsequently, we will try to see if there is a crown or coronoradicular fracture and identify in this decision-making tree the correct diagnosis that will lead to the development of the appropriate treatment plan. In any case in which there is loss of tooth structure, thorough exploration should be added, even with the realization of radiographs of soft tissues where this lost tooth material could have been located [18].

In this article, we will present the clinical case of an elderly patient who suffered a dental trauma as the result of a fall and who received an initial inadequate diagnosis, which resulted in a lip complication with pain and edema.

## 2. Clinical Case

This case report complies with the Helsinki Declaration, and the patient signed an informed consent form.

A 79-year-old female patient presented to the office with edema and pain in the lower lip. The medical history revealed diabetes, Sjögren’s syndrome, and leukoplakia.

As part of her dental history, the patient explained that she suffered a fall six months ago while strolling on the street, resulting in a fracture of the upper incisors and greater mobility of the lower incisors.

On the day of the fall, the patient was examined by a dentist in the emergency room. She was diagnosed with a simple crown fracture of the right upper central incisor and the right upper lateral incisor without any pulpal involvement and with laceration of the lower lip (Figure 1A–C). In the emergency examination, pulp vitality tests and periapical X-ray of the upper and lower incisors were performed to rule out root fractures or other injuries (Figure 1D–F). She was prescribed a soft diet and analgesics in case of pain, and she was advised to have a follow-up with her usual dentist.

However, the patient did not visit her usual dentist until six months after the accident, due to pain and inflammation of the lower lip.

On current examination, an edematous lower lip was observed (Figure 2A), and a hardened painful area was detected on palpation. An X-ray of the lower lip revealed a radiopaque mass compatible with a fragment of dental tissue (Figure 2B).

A surgical excision of the fragment facilitated by a minimal incision on the lip was indicated, and the fragment was subsequently extracted (Figure 2C,D).

After wound healing and a resolution of the problem, the fractured incisors were restored (Figure 2E,F), and the patient was informed of the need to maintain good oral health, recommending periodontal treatment and subsequent check-ups to monitor the evolution.

## 3. Discussion

The correct attitude in TDI situations is not well known in both the health-care sector and among dentists [19,20]. Any patient who has suffered a TDI should be considered an emergency case and should be treated immediately, not only clinically to alleviate symptoms and improve the patient’s quality of life but also to improve the prognosis of the injury. Perhaps when comparing dental injuries with other types of corporal trauma, the severity can be considered a much lesser entity; however, it must be taken into account that the recovery capacity of the teeth is much lower after an injury, so the establishment of the correct treatment is extremely important [21,22].

Making a mistake can have important consequences for the recovery of the patient affected by the event as well as the maintenance of teeth, healthy pulp, esthetics, speech, and mastication. Various associations, societies, and professional colleges offer advice and guidance on how to act when faced with a TDI, and it is advisable to access them to gain and improve our knowledge. For example, the International Association of Dental Traumatology (IADT) has developed protocols that are periodically updated in order to help dental professionals keep up to date due to the great complexity of establishing a correct, precise, and timely treatment for dental traumatic injuries [23].

The standards of treatment for maxillofacial injuries are comparable for all ages. In any case, geriatric patients with facial breaks require fitting measures for their administration since, in spite of the same etiology, by and large, within the elderly, the breaks are more serious [17,24]. It is exceptionally important to inquire about assessing systemic comorbidities, such as cardiovascular illness, in elderly patients who have endured facial injury [13], since frequently these can have an impact over the course of treatment.

It is amazingly imperative that a multidisciplinary care framework is marked out and mindful of geriatric patients, in which the realization of the legitimate administration of facial injury can bring, beyond the financial benefits to wellbeing frameworks, to improve the patient’s quality of life [13,17,24]. According to Brucoli et al. (2020), nearly half of the cases of elderly patients with a facial injury require a specialized interview prior to surgery due to the closeness of other systemic co-morbidities or other related wounds, deferring the completion of surgery for 72 h after the affirmation [13].

An initial clinical investigation in the earliest stages following a traumatic episode is vital, including a thorough oral examination and in-depth inspections, palpation, and percussion, in order to give a correct diagnosis. Only by following an established protocol will we be able to correctly detect the clinical situations discussed here.

Since 1985, standard questionnaires have been published to emphasize the importance of carrying out exhaustive diagnoses and recommendations, which are still valid today [25]. In order for the anamnesis and clinical examination to be as quick and exhaustive as possible, and with the purpose of not forgetting any important aspects, it is convenient to systematically use a specific clinical history form. This will not only facilitate the planning of the treatment that we decide to carry out when we reach an adequate diagnosis, but it can also be useful from a legal point of view, bearing in mind that dental traumatisms are often the result of accidents or blows that may require reports from insurance companies or any other type of medical–legal claims [26].

Following this systematic approach, after a series of questions about the trauma (asking when, how, the cause, and if the patient has received previous treatment), a brief neurological examination of the patient will be carried out, followed not only by the mere sight of the traumatized tooth but also by an extraoral analysis where the exploration and palpation of the head and temporomandibular joint will be examined. The soft tissues and skin will be checked for the existence of tumefaction, contusion, abrasion, ceration, or penetrating wounds where tooth fragments or foreign bodies could be lodged. The most probable mechanism of the impact of this type of foreign body inside the soft tissues as a result of trauma is that the tooth, having penetrated this tissue, fractures while emerging from the skin and collides with a hard object, so that in the fracture, it ends up lodged inside these tissues. In these cases, the diagnosis is very difficult because inflammation and edematization of the tissues increase their hardness and can alter the perception of palpation. That is why it is necessary and absolutely essential before performing a suture in this type of trauma to perform conventional radiographs of the soft tissues with the lacerating wound to rule out or confirm the presence of foreign bodies [22,26].

For a correct exploration of soft tissue wounds, we should begin by cleaning the affected surfaces with gauze soaked in a saline solution, removing all the embedded material resulting from the trauma, and also avoiding the appearance of any possible tattooed areas. In the event that superficial washing and brushing of the lesion with a soft brush cannot remove all the encrusted material, we can, after local anesthesia of the patient, use a magnification of vision for the use of fine scalpel blades (e.g., number 11) to remove this encrusted debris. The use of antibiotic or antiseptic ointments or dressings for wound decontamination has also been described [22,26].

Any perioral soft issue can be the location of hard dental matter. Our bibliography highlights the most frequent location of hard tissue inclusions in the lower lips, a consequence of fractures in the central upper incisor region [27,28], Nagaveni and Umashankara [27] reported a case of the inclusion of a dental fragment in the lower lip of a ten-year-old child, which remained there for a period of 10 months, as occurred in our case. However, there are exceptional cases, such as the one described by Santana Santos et al. [29], with the inclusion of an upper premolar and a lower incisor, respectively, in the tongues of two patients who suffered a firearm assault. The impact of the bullet caused dental trauma, with the expulsion of dental matter acting as a second bullet.

Most reported cases correspond to the first decades of life [30]. Pradeep Kumar, R. [25] reports the case of a female patient aged 29, and Santana Santos et al. [29] present two cases of male patients aged 35 and 34 years, respectively. We have not found such published cases of elderly patients.

The treatment requires the removal of the included fragment and restoration of the fractured tooth wherever possible. On occasion, we can reposition the fragment using appropriate adhesive techniques. It is necessary for the fragment to fit perfectly, but this procedure should be performed soon after the event to ensure that no deterioration occurs to the surface where the fragment is to be adhered, although there have been cases where this procedure has taken place two months after the initial trauma [28].

On other occasions, there is not just a single fragment but many smaller fragments in the soft tissue [31,32], which makes removal difficult and prevents any repositioning of the fragment. The likelihood of complications from fibrosis, infection, and displacement of hard tissue increases if there is a delay in the extraction of the fragment [30,33]; therefore, an accurate diagnosis is needed as quickly as possible after the traumatic event [32]. However, on several occasions, the patient has a compromised general condition, and the attention of medical services is prioritized. In such cases, a delay in dental treatment facilitates the presence of post-traumatic complications.

The prevention of injuries and the need for adequate dental treatment in geriatrics are important because this group of people has special needs owing to either their medical or psychological situation. The overall health of the patient cannot be conceived without adequate oral health.

The dental team, family members and/or caregivers, and the patients themselves must treat and attend dental trauma check-ups in geriatrics with the same vigor and care as that of a young patient.

## 4. Conclusions

The population of geriatric patients is steadily increasing in our society, and a higher frequency of falls has been observed in this age group, affecting the maxillofacial region, with consequent dental trauma on many occasions. Given the increasing incidence of this type of trauma, an adequate diagnostic and treatment protocol should be followed according to the specific case to avoid easily solvable problems.

Therefore, as professionals, we must pay special attention to the prevention and treatment of this problem, insisting on the importance of routine investigation in emergency visits for dental trauma, which includes a meticulous examination of the soft tissues accompanied by a radiographic examination, when the inclusion of foreign bodies is suspected.

## Figures and Tables

**Figure 1 diagnostics-13-03021-f001:**
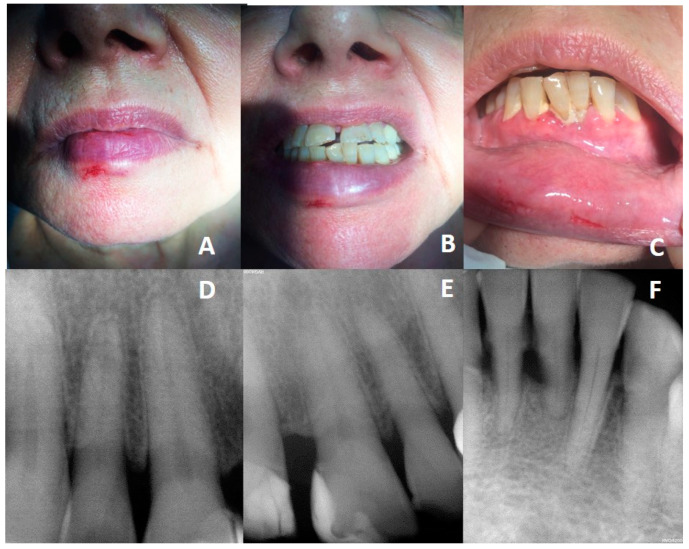
Images of the day of the accident. (**A**): Facial injuries. (**B**): Upper incisor fracture. (**C**): Laceration of the lower lip. (**D**,**E**): Periapical X-rays of the upper incisors. (**F**): Periapical X-ray of the lower incisors.

**Figure 2 diagnostics-13-03021-f002:**
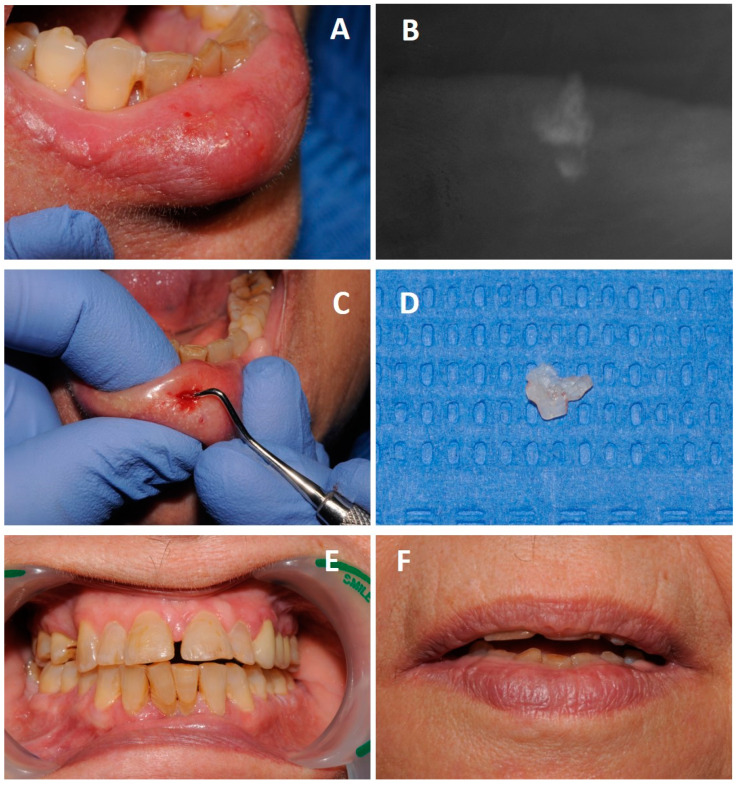
Images taken six months after the accident. (**A**): Inflammation of the lower lip. (**B**): Intraoral X-ray of the lip. (**C**): Surgical excision of the fragment. (**D**): Fragment of dental tissue extracted from the lip. (**E**): Clinical image after resolution of the case. (**F**): Clinical Image in a control three years later.

## Data Availability

All data will be sent by corresponding authors if it is required.

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
