# Peer review of "Complications Arising from Dental Trauma Incurred from Falls Involving Geriatric Patients: A Case Report"

_diagnostics, 2023, doi:10.3390/diagnostics13193021_

Round 1
Reviewer 1 Report
Comments and Suggestions for Authors
The title does not accurately describe that this paper is a case study report.
The authors may be trying to do too much in the paper rather than concentrating on thoroughly describing the case at hand. I would suggest that the authors look at other case reports published in high impact journals to help guide them.
I would suggest that the authors have to make it clear as to why this case study is beneficial to the readership i.e. the purpose of the case study and then appropriately provide background information followed by specific details of the case study and ending with lessons learnt (perhaps a checklist or flowchart to help clinicians better assess & manage soft tissue injuries associated with dental trauma).
There needs to be some parameters set for the literature review, such as, some key terms and databases used in the search.
I suggest the authors also include: Manteen N. Facial trauma in geriatric patients: epidemiology and complications (2021) The Health Review.
The conclusion is too broad and is not supported by the narrow remit of the case study.
Comments on the Quality of English Language
The manuscript would substantially benefit from review by an English language editing service as there are considerable errors, beginning from the title ( edlers patients) and throughout the manuscript.
Reviewer 2 Report
Comments and Suggestions for Authors
This is an interesting case report presenting a case in which the fractured tooth was not detected in patient's lower lip and caused inflammation. A meticulous examination and radiographic examination of the soft tissue was recommended. This case report provided more information of elderly maxillofacial trauma, and will help improving clinical examinations in the dental practice for elderlies.
Comments on the Quality of English LanguageNA
Round 2
Reviewer 1 Report
Comments and Suggestions for Authors
The authors have now improved the manuscript by adopting suggested changes.
Comments on the Quality of English LanguageMinor grammatical errors need to be corrected.